# Cross-reactive neutralizing human survivor monoclonal antibody BDBV223 targets the ebolavirus stalk

Liam B. King[1,9], Brandyn R. West[1,9], Crystal L. Moyer[1], Pavlo Gilchuk[2], Andrew Flyak[3], Philipp A. Ilinykh[4,5], Robin Bombardi[2], Sean Hui [1], Kai Huang [4,5], Alexander Bukreyev [4,5,6], James E. Crowe Jr. [2,3] & Erica Ollmann Saphire [1,7,8]

Three *Ebolavirus* genus viruses cause lethal disease and lack targeted therapeutics: Ebola virus, Sudan virus and Bundibugyo virus. Monoclonal antibody (mAb) cocktails against the surface glycoprotein (GP) present a potential therapeutic strategy. Here we report two crystal structures of the antibody BDBV223, alone and complexed with its GP2 stalk epitope, an interesting site for therapeutic/vaccine design due to its high sequence conservation among ebolaviruses. BDBV223, identified in a human survivor of Bundibugyo virus disease, neutralizes both Bundibugyo virus and Ebola virus, but not Sudan virus. Importantly, the structure suggests that BDBV223 binding interferes with both the trimeric bundle assembly of GP and the viral membrane by stabilizing a conformation in which the monomers are separated by GP lifting or bending. Targeted mutagenesis of BDBV223 to enhance SUDV GP recognition indicates that additional determinants of antibody binding likely lie outside the visualized interactions, and perhaps involve quaternary assembly or membrane-interacting regions.

[1] Department of Immunology and Microbiology, The Scripps Research Institute, La Jolla, CA 92037, USA. [2] Vanderbilt Vaccine Center, Vanderbilt University Medical Center, Nashville, TN 37232, USA. [3] Departments of Pediatrics, Pathology, and Microbiology and Immunology, Vanderbilt University Medical Center, Nashville, TN 37232, USA. [4] Department of Pathology, University of Texas Medical Branch, Galveston, TX 77555, USA. [5] Galveston National Laboratory, Galveston, TX 77555, USA. [6] Department of Microbiology & Immunology, University of Texas Medical Branch, Galveston, TX 77555, USA. [7] Skaggs Institute for Chemical Biology, The Scripps Research Institute, La Jolla, CA 92037, USA. [8] La Jolla Institute for Immunology La Jolla, CA 92037, USA. [9] These authors contributed equally: Liam B. King, Brandyn R. West. Correspondence and requests for materials should be addressed to E.O.S. (email: erica@lji.org)

Filoviruses cause sporadic and unpredictable outbreaks of human disease. The two homopathogenic genera within the family are *Ebolavirus* [including Ebola virus (EBOV), Sudan virus (SUDV), Bundibugyo virus (BDBV), Taï Forest virus, and Reston virus] and *Marburgvirus* [containing Marburg virus (MARV), and Ravn virus]. During the 42-year history of Ebola virus Disease (EVD) outbreaks, case fatality rates have ranged from 25 to 90% depending on the infecting virus, location, and other factors. The 2013-2016 EVD epidemic in West Africa occurred in a location not previously known to harbor EBOV, and ultimately infected and killed over 28,000 and 11,000 people, respectively. Currently, there is no approved therapeutic to treat EVD.

Ebolaviruses are enveloped ssRNA viruses which express eight proteins. The trimeric spike glycoprotein (GP) on the viral surface is responsible for attachment and entry into the target cell. Due to its exposed nature and critical role in the viral life cycle, GP is an attractive target for drug and therapeutic design. In the host cell, newly synthesized GP is cleaved by the host protease furin to yield two subunits, GP1 and GP2, which remain linked by a single disulfide bond[1,2]. GP1 contains the host receptor-binding site, the glycan cap, and the flexible, heavily glycosylated mucin-like domain[3]. GP2 contains the N-terminal peptide, the internal fusion loop, two consecutive heptad repeat regions (HR1 and HR2), the membrane proximal external region (MPER), and the C-terminal transmembrane domain[3,4]. HR2 is a largely alpha-helical section of protein, also termed the stalk, that connects the GP core to the viral membrane.

Filoviruses are internalized into target cells by macropinocytosis[5–8]. Upon entering the endosome, ebolavirus GP is processed by endosomal cysteine cathepsins B and L[9,10] that cleave the glycan cap and mucin-like domain from the GP surface to expose the receptor-binding site[11–13]. After receptor binding, GP2 undergoes conformational rearrangements to form a six-helix bundle that drives fusion of the virus and host membranes through mechanisms that are not well understood[14,15].

The stalk region of GP2 connects the globular body of GP to the viral or cell membrane and is the most C-terminal section of GP that has been visualized to high resolution[16]. This region is of interest for therapeutic/vaccine design due to its relatively high amino acid sequence conservation among the ebolaviruses: 71% identical by primary amino acid sequence among five ebolaviruses, but 90% identical among EBOV, BDBV, and SUDV, the three ebolaviruses most frequently linked to human disease.

The antibody BDBV223 was identified in a human survivor of the 2007 BDBV outbreak in Uganda and targets the GP2 stalk[17]. Although it was elicited during BDBV infection, it also cross-reacts to, neutralizes, and protects mice and guinea pigs against heterologous EBOV[18]. EBOV and BDBV GP differ in the stalk region at two sites, V631I and T634P (Supplementary Fig. 1), but apparently neither polymorphism matters for binding or neutralization of EBOV[17]. SUDV and BDBV GP, however, differ in this region at two other sites, D624N and K633N. BDBV223 binds to recombinant SUDV, but is unable to neutralize SUDV[18]. Among these substitutions, D624N is key: a D624N mutation abrogates BDBV223 binding, while a K633N substitution retains BDBV223 binding[17].

Here we describe two crystal structures of BDBV223: one alone, and one in complex with a synthetic peptide corresponding to its BDBV GP stalk epitope, at 2.0 and 3.7 Å resolution, respectively. Modeling of the antibody-GP2 stalk complex into a map of a transmembrane GP assembly reveals that binding of BDBV223 to GP likely interferes with the GP quaternary assembly and captures the bound stalk in a position that is pulled away from the viral membrane.

## Results

**Structures of unbound and GP peptide-bound BDBV223.** BDBV223 Fab was co-crystallized with a synthetic peptide [620]TDKIDQIIHDFIDKPL[635] representing residues 620–635 of the BDBV GP2 stalk epitope[17]. Crystals of the BDBV223-GP2 stalk complex diffracted at SSRL 12–2 to 3.7 Å resolution, and crystals of the unbound Fab fragment diffracted at APS 23-ID-D to 2.0 Å (Supplementary Tables 1 and 2). Two Fab-peptide complexes are in the asymmetric unit of the complex, and one Fab is in the asymmetric unit of the apo-Fab structure (Fig. 1). Both copies of the Fab-peptide complex are essentially identical in structure, and CDR H3 is ordered and visible in its entirety in both copies. However, in the apo-Fab structure, seven residues of CDR H3 (R100-S106) are disordered, suggesting inherent flexibility in this region in the absence of its bound GP epitope.

**Induced fit changes in heavy chain CDRs.** Sequence analysis of the variable regions of mAb BDBV223 identified IGHV4-34*01 and IGKV3-20*01 as germline precursors for its heavy and light chain (Supplementary Fig. 2 and Supplementary Table 3). The level of somatic hypermutation of these regions was 15.5% and 14.7%, respectively. Although the arrangement of light chain CDRs of the BDBV223 Fab is similar in the apo- and peptide-bound structures, positions of each of the CDRs H1, H2, and H3 significantly shift upon binding the BDBV GP stalk epitope (Fig. 2 and Supplementary Fig. 3). CDR H1 translates ~6 Å towards the N terminus of the GP peptide when bound (measured at Cα of T31) (Fig. 2a–c). CDR H2 shifts 1.6 Å when in complex with peptide, and this shift is also accompanied by a rotameric change in Y53 that repositions its phenolic oxygen 7.4 Å towards the antigen (Fig. 2d–f). The most prominent rearrangement upon BDBV223 Fab binding to the GP stalk peptide, however, occurs in CDR H3 (Fig. 2g–i). The ordered portion of CDR H3 in the unbound structure adopts a different trajectory compared with the peptide-bound structure. Upon peptide binding, the loop re-orients towards the light chain by ~5.6 Å at the base (measured from the Cα of I99; the tip is only ordered in the complex) and the CDR H3 stem loses its β-strand character and becomes kinked. These structural shifts illustrate a high degree of conformational flexibility in BDBV223 HCDRs that likely influences the binding energy and pathways by which antibodies such as BDBV223 are elicited against this region. These structural rearrangements also point to the inherent challenges in prediction of Fab binding modes from non-complexed antibody structures alone.

**CDR-peptide interactions.** The GP2 stalk lies in the C-terminal region of the protein, just after residue C609, which mediates the disulfide bond to GP1. Current structural information indicates that the stalk forms an alpha helix that uncoils near its C terminus as it transitions into the MPER. The final C-terminal residue visible in previous ebolavirus structures is D632[16]. The peptide in this BDBV223 complex structure contains GP2 residues 620–635 and maintains its alpha-helical structure until residue I627.

Heavy chain CDRs interact across the entire span of the GP peptide. Although CDR H3 residues 100–106 are disordered in the unbound BDBV223 structure, they are ordered in the complex and form the largest contribution to the paratope in the bound structure. CDR H3 residues R98, R100, A104, Y105, and S106 each interact with the GP helical epitope. CDR H3 residue R100 is located 3.8 Å away from GP residue D624 (aspartate in both EBOV and BDBV) and these two residues likely form a salt bridge (Fig. 3 and Supplementary Fig. 4). Versions of BDBV223 bearing R100A and R100W point

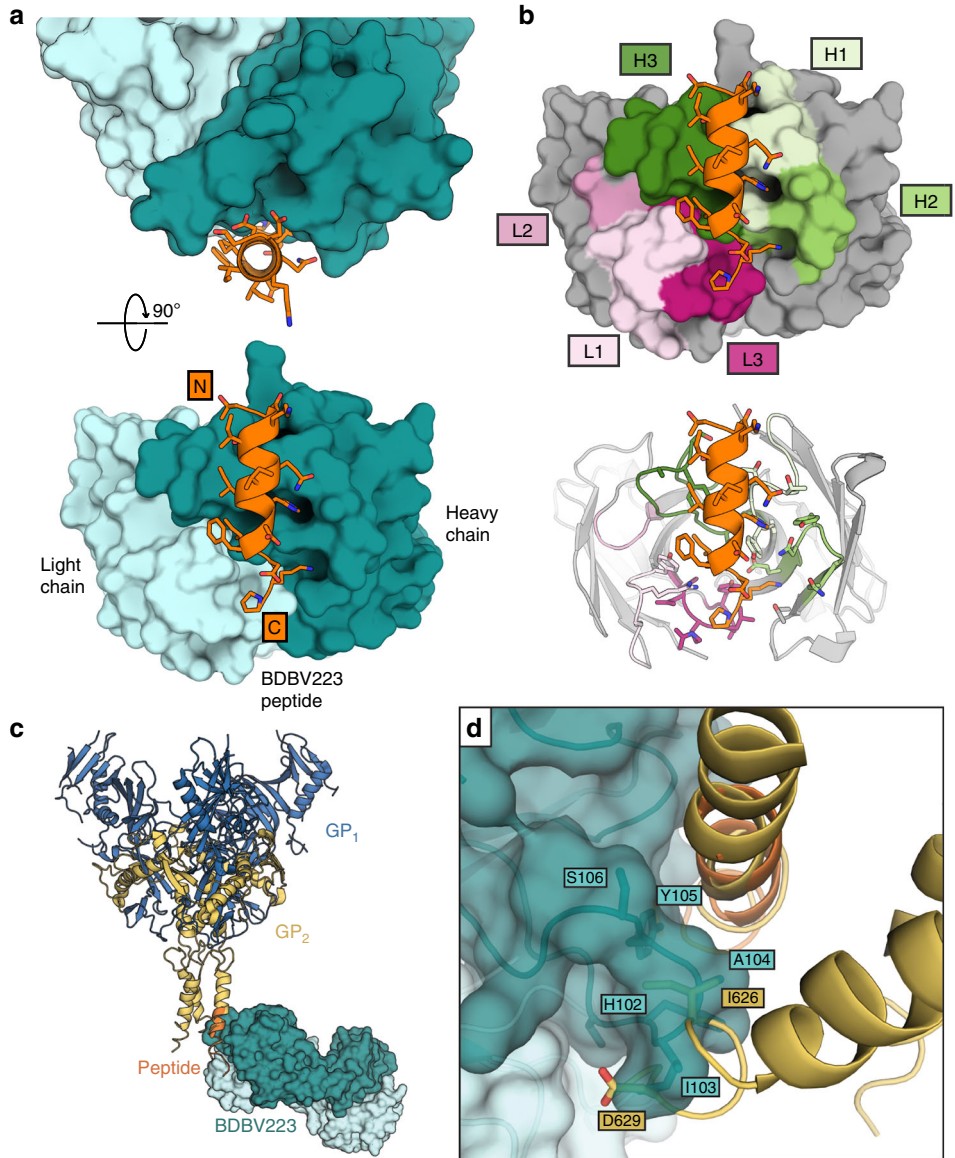

**Fig. 1** Structure of BDBV223-stalk peptide complex. **a** Top and front view of BDBV223 bound to the stalk peptide (orange). The heavy and light chains of the antibody are dark and light teal, respectively. **b** View of BDBV223 highlighting the molecular surface contributed by each CDR in the paratope. **c** Alignment of the crystallized GP2 peptide (orange) to the full-length EBOV GP structure (yellow; PDB: 5JQ7)[16] illustrates how the BDBV223 Fab anchors to the visible C-terminus of the ectodomain portion of the GP2 stalk. **d** The structure by which BDBV223 anchors to one GP2 protomer, however, is incompatible with the close trimeric bundle arrangement observed in the unbound GP trimer structure, as the bound antibody, particularly its CDR H3, sterically clashes with residues 626–629 of the neighboring GP monomer. Color Scheme: RGB

mutations do not bind to GP. Other mutations in this region are better tolerated (Supplementary Figs. 5 and 6). In addition to forming a probable salt bridge with CDR H3 residue R100, GP residue D624 also forms an H-bond with CDR H3 residue S106. SUDV bears N instead of D at position 624, but also requires S106 for binding; SUDV GP is not recognized by S106A or S106N-bearing antibody (Supplementary Fig. 5). CDR H3 Y105 forms van der Waals interactions with GP residue I627. The other three significant CDR H3 residues that form the BDBV223 paratope include R98, H102, and A104, each of which participate in van der Waals interactions with I627, F630 and I631 of the GP peptide (Fig. 3).

Among the three consecutive threonines of CDR H1 (T30, T31, and T32), the main chain carbonyls of T30 and T31 contact the H628 side chain of the GP peptide. However, the most important CDR H1 residue involved in GP peptide interaction is Y33, which

appears to engage in a π-stacking interaction with H628. In this interaction, the Y33 side chain appears to insert into the GP helix and destabilize the secondary structure of the GP peptide (Fig. 3c). The main chain carbonyl of GP residue H628 also hydrogen bonds with Y33. However, BDBV223 antibody bearing a Y33F mutation maintains binding to BDBV and EBOV GP, suggesting that for these viruses, the aromatic ring stacking is the key interaction. Y33F-bearing antibody, however, does not bind to SUDV GP (Supplementary Fig. 5). In CDR H2, both N52 and Y53 also appear to interact with H628 by van der Waals forces.

The BDBV223 light chain interacts with the C-terminal section of the GP peptide. CDR L1 interfaces with the C-terminal end of the peptide through R31 and Y33 (Fig. 3d). CDR L3 residues R94, L95, and the main-chain carbonyl of D93 contact P634 of the GP peptide. Notably, the only escape mutation yet raised in the binding epitope of BDBV223 is P634H (Fig. 3e)[17]. Based on this

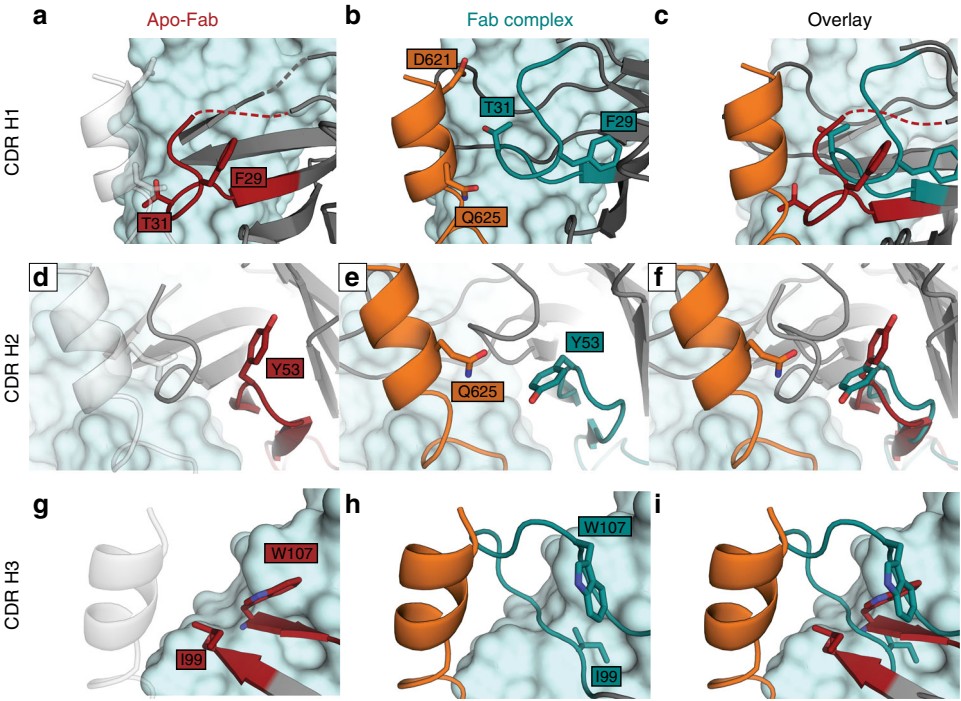

**Fig. 2** Heavy chain CDR movements upon binding the peptide. The heavy chain CDRs of BDBV223 undergo induced fit rearrangement upon binding the peptide. **a–c** CDR H1 maintains a similar structure overall, but shifts towards the N-terminus of the peptide. **a** Unbound CDR H1 is illustrated in red. **b** Bound CDR H1 is illustrated in cyan. **c** Superimposition indicates the repositioning of CDR H1 upon binding of the GP2 peptide. **d–f** CDR H2 does not have a significant main chain shift, but shows a significant rotameric change in Y33 towards GP2, where the helical character breaks down. **g–i** CDR H3 demonstrates the largest induced fit rearrangement, with significant shifts in the positions of anchor point residues I99 and W107 and ordering of contact residues 100–106 only upon GP binding. The peptide is colored orange in the complex and is transparent in the Apo-Fab structure. Changes in the light chain are minimal. Color Scheme: RGB

structure, it appears that this mutation would clash primarily with R94 of CDR-L3 (electrostatic representation shown in Supplementary Fig. 7).

**Likely interference with the trimeric interface**. In unbound GP structures, the three stalk α-helices in a GP trimer maintain a close association in a trimeric bundle[11,16,19]. However, in this crystal structure, BDBV223 wraps its CDR H3 around the α-helix, forming multiple van der Waals contacts, hydrogen bonds, and a likely salt bridge (there are no crystal contacts that induce this conformation). Superimposition of this Fab-GP stalk peptide structure onto GP trimer structures indicates that CDR H3 of BDBV223 would clash with an adjacent GP monomer in the trimeric stalk bundle (Fig. 1). Although the high-resolution unbound EBOV GP was expressed fused to a fibritin fold-on domain, which could theoretically induce a non-native conformation, we note that a native marburgvirus structure solved without the use of a trimerization domain adopts a similar tight trimeric bundle (Supplementary Fig. 8)[19]. Binding of BDBV223 would likely destabilize this tight bundle if it occurs in native EBOV and BDBV GP, and may impede six-helix bundle formation of GP2 during fusion[14,15].

An N-linked glycosylation sequon is encoded in the GP2 stalk of all ebolaviruses and marburgviruses (N618 in ebolavirus GP and N619 in marburgvirus GP)[16,19], and an attached glycan at this site has been visualized for EBOV, but not marburgvirus GP[16,19]. N618 lies in the center of the GP2 stalk and the presence of a glycan attached in that center may generally interfere with antibody recognition of this conserved stretch of polypeptide. BDBV223, however, binds just "underneath" this glycan, on the lower, C-terminal half of the stalk. The peptide representing the epitope of BDBV223, GP residues 620–635, begins approximately

half of a helical turn below the glycan, and the bound BDBV223 antibody is positioned 90° around the helical stalk from the glycan attachment site.

**Differences in SUDV**. BDBV223 neutralizes BDBV and EBOV but fails to neutralize SUDV. Residues 620–634 of the BDBV223 peptide epitope are visible in the crystal structure (residue 635 is disordered in the crystal lattice). There are two side chains in this epitope that differ between BDBV and EBOV GP and two others that differ between BDBV and SUDV GP. More specifically, BDBV and EBOV GP differ at I631V and P634T, neither of which appear to be detrimental to cross-neutralization. BDBV GP I631, visualized in this structure, forms hydrophobic contacts between CDRs L1 and H3. Val 631 of EBOV GP may form similar hydrophobic interactions in a complex with BDBV223. P634 forms a van der Waals contact with CDR L3 that could also be adopted by the Thr of EBOV. Differences in BDBV vs. SUDV GP include K633N and D624N. Contacts by K633 are mediated by the carbon atoms of the lysine side chain, and it has been previously observed that a K633N mutation does not abrogate BDBV223 binding[17]. At position 624, however, SUDV encodes an Asn instead of an Asp, a substitution that is known to disrupt binding by the BDBV223 antibody[17]. In this crystal structure, D624, encoded by BDBV and EBOV, is observed to likely form a salt bridge with R100, a chemical interaction that is inaccessible to the non-ionizable N624 of SUDV. Further, the Asp-containing native BDBV GP stalk has a complementary electrostatic surface profile to the BDBV223 paratope, in which the acidic D624-encoding surface interacts with the basic R100-containing surface of the heavy chain (Fig. 3). The D624N polymorphism (BDBV GP to SUDV GP) would decrease the acidic nature of the N-terminal region of the stalk.

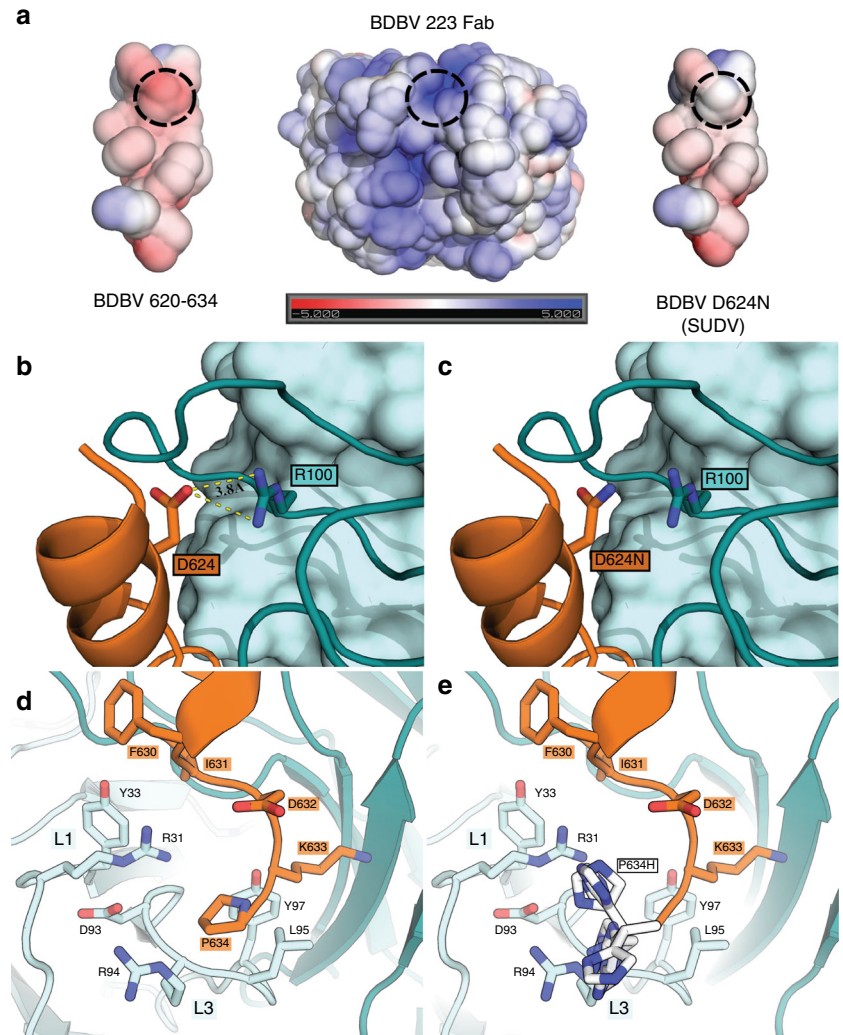

**Fig. 3** Modeling of the D624N BDBV/SUDV polymorphism. **a** Left, electrostatic surface representation of the interfacing surfaces of the BDBV GP stalk (620–635). Right, model of the BDBV GP peptide bearing a D624N mutation. Center, the binding cleft of BDBV223 Fab. Replacing the acidic Asp residue for an Asn reduces the predicted electrostatic attraction at the N-terminus of the peptide. **b** The potential salt bridge formed between D624 and R100 would be abolished in the case of an Asn substitution, as occurs in SUDV. **c** CDRs H1 and H2 largely interact with H628, present in all three pathogenic ebolaviruses (EBOV, BDBV, and SUDV), with CDR H1 residue Y33 forming an apparent pi-stacking interaction with H628. **d** CDRs L1 and L3 interact with the C terminus of the stalk peptide and form close contacts with I631 and P634. **e** Based on this model, the previously described escape mutant P634H[17] would likely clash most prominently with R94. Color Scheme: RGB

A series of single point mutations in BDBV223 were evaluated to (a) probe the importance of antibody side chains in the vicinity of D624 and (b) determine if additional hydrophobic contacts could make up for the loss of the salt bridge and improve binding and neutralization of N624-bearing SUDV. Heavy chain Ser 106, which forms an H-bond to the carbonyl oxygen of GP D624 was mutated to Ala. Heavy chain Ile 103, which appears to disrupt the trimeric interface of GP, was mutated to Ala and acidic Glu. Two residues were mutated to Phe: heavy chain Ala 104, oriented toward I627, and Tyr 33 which forms π-stacking interactions with H628 of GP2 (Supplementary Fig. 5). None of these antibody mutations abrogated BDBV neutralization, but all mutations reduced EBOV neutralization to some degree (Supplementary Fig. 6). Three of the mutations, A104F, I103A, and I103E, were able to bind SUDV GP (Supplementary Fig. 5), but none of the mutants neutralized SUDV GP-bearing virions (Supplementary Fig. 6). Continued failure to neutralize SUDV suggests that the likely salt bridge to the ionizable Asp of BDBV and EBOV could be a critical determinant of neutralization that cannot be replaced. The robustness of BDBV neutralization, sensitivity of EBOV neutralization, and inability to achieve SUDV neutralization despite improving binding to SUDV together suggest that additional determinants of neutralization capacity lie outside the visualized 620–634 peptide epitope, perhaps involving differences among the viruses in quaternary assembly, glycan incorporation, membrane interaction, or another feature not recapitulated in the visualized Fab-polypeptide interaction.

## Discussion

There are multiple known ebolaviruses that are antigenically distinct. At least three of these viruses (EBOV, BDBV, and SUDV) have been linked to numerous outbreaks among humans. All investigational therapies and vaccines on the WHO list for the 2018 outbreak in DRC, however, are specific only for EBOV, with little to no activity against the other ebolaviruses. A primary goal for the field is the characterization and development of more broadly cross-reactive antibody therapeutics and vaccines that are protective against whichever ebolavirus may emerge or re-emerge to cause human disease. BDBV223, isolated from a B cell of a

human survivor of the BDBV outbreak in Uganda in 2007, potently neutralizes both BDBV and EBOV[18]. BDBV223 binds a GP epitope thus far untargeted by neutralizing antibody therapeutics mobilized for human use, HR2 of the GP2 stalk.

Here we compare the crystal structures of BDBV223 alone and in complex with a synthetic peptide, GP residues 620–635, representing its GP2 epitope in BDBV. In the crystal structures we observe significant induced fit in which each of the heavy chain CDRs shifts in position and rotates a key side chain to wrap about the alpha helix of the GP2 stalk monomer. BDBV223 binds to an exposed polypeptide in HR2, just below the conserved N619-linked glycan, and just above the viral or cell membrane to which GP is anchored.

Visible in the crystal structure are all but one of the residues of the GP2 peptide encompassing residues 620–635. The peptide extends three residues beyond the previously observed C-terminus in a high-resolution EBOV GP structure[16]. Interestingly, superimposition of the structure of the BDBV223-peptide epitope complex onto the structure of unbound trimeric EBOV GP indicates that binding of BDBV223 is incompatible with the presence of the trimeric stalk bundle of the unbound GP. Wrapping of CDR H3 about a GP2 stalk protomer would separate GP2 from its neighboring protomers in the trimer (Fig. 1). The BDBV223 epitope is accessible in the post-fusion six-helix bundle conformation of GP2[14], but binding of IgG3 BDBV223 could interfere with rearrangement of GP2 into this form (Supplementary Fig. 9).

Another steric clash is apparent in modeling studies as well. Alignment of the BDBV223-crystal structure to a cryo-EM reconstruction of unbound EBOV GP on the viral membrane surface[20] indicates that the angle of approach of BDBV223 is incompatible with the position of its epitope in relation to the membrane. The GP stalk must either lift or bend out of this position in order for BDBV223 to bind. Steric hindrance would be even more pronounced for the natural intact IgG3 form of BDBV223 in which it was originally isolated from survivor cells (Fig. 4). Further, low-resolution negative-stain EM class averages illustrate heterogeneous angles of approach of BDBV223 beneath the GP core, supporting the idea of displaced GP stalks upon binding[17,21].

The crystal structure reveals that CDR H3 residue R100 likely forms a salt bridge with D624, which is shared between BDBV and EBOV. SUDV, however, encodes N624 at this position, a residue that is non-ionizable at physiologic pH and is unable to form the same salt bridge. Note that pH does not impact binding of BDBV223 to BDBV or EBOV GP (Supplementary Fig. 10). Interestingly, mutagenesis and engineering based on the protein–protein interactions of the Fab-GP2 peptide complex had different effects on different viruses: neutralization of BDBV was robust for all antibody substitutions, neutralization of EBOV was sensitive and diminished upon antibody side chain substitution, and although several mutants bound SUDV, neutralization of SUDV was unachievable. We expect that the likely salt bridge to D624 encoded by BDBV and EBOV, but missing in SUDV, is required for neutralization and/or that there are determinants of binding and neutralization for this antibody that lie outside of its footprint on a GP monomer. One possibility may be different degrees of mobility in the MPER or transmembrane domains, or differences elsewhere in the GP core such as glycan structure or quaternary assembly. Another possibility is that the different electrostatics at the SUDV GP core cause electrostatic repulsion in trimer assembly and stability compared to EBOV[22]. The CDR H3 of antibody 4E10 against the MPER of HIV-1 also recognizes a linear, helical MPER epitope which appears to be determined or constrained by the viral membrane, and requires lifting or bending of the epitope out of the membrane. Perhaps BDBV223,

like 4E10, also requires an antigenic structure that is more complex than that of the monomeric linear helical epitope itself in order to achieve full binding[23].

Cumulative evidence indicates that our understanding of the organization of GP on the infected cell or the authentic virus is incomplete. Further exploration beyond the GP ectodomain is required to understand activity of anti-stalk antibodies that are elicited by natural infection or vaccination, and how they could be developed into therapeutics.

## Methods

**Expression/purification.** Human hybridoma cells secreting BDBV223 were expanded in post-fusion medium[24]. Briefly, cells were fused with HMMA2.5 myeloma cells, and then hybridomas producing BDBV223 were cloned by two rounds of limiting dilution and by single-cell fluorescence-activated cell sorting. Once cloned, hyridoma cells were expanded in post-fusion medium (ClonaCell-HY Medium E, STEMCELL Technologies #03805) until 50% confluent in 75-cm$^2$ flasks (Corning #430641). For production of BDBV223, cells from one 75-cm$^2$ flask were collected with a cell scraper and expanded to four 225-cm$^2$ flasks (Corning #431082) in serum-free medium (Hybridoma-SFM, Gibco #12045–076). After 21 days, the supernate was clarified by centrifugation and sterile filtered using a 0.2-μm pore size filter device. HiTrap Protein G or HiTrap MabSelectSure columns (GE Healthcare Life Sciences #17040501 and #11003494 respectively) were used to purify BDBV223 from filtered supernate. Fab fragments were generated from purified IgG through digestion with 4% papain for 6 h followed by purification over HiTrap KappaSelect column (GE Healthcare). The BDBV 620–635 peptide was generated synthetically (GeneScript).

**Crystallization.** Purified BDBV223 Fab was concentrated to 13.5 mg/mL for crystallization. The unbound Fab (apo) was crystallized via sitting drop vapor diffusion at room temperature using 0.2 μL of protein (13.5 mg/mL of protein, 150 mM Tris-HCl pH 7.5, 10 mM NaCl) plus 0.2 μL 80 mM sodium citrate tribasic/hydrochloric acid pH 5.6, 20% PEG 300, 200 mM ammonium sulfate, and 10% glycerol. To form complexes, Fab was combined with a 10-fold molar excess of peptide and incubated at 4℃ for 48 h. The Fab-peptide complex was crystallized in 0.2 M magnesium acetate, 0.1 M sodium cacodylate: HCl, pH 6.5, 20% (w/v) PEG 8000. Fab-peptide complex crystals were soaked in mother liquor+20% glycerol for cryoprotection. Crystals were harvested and flash-cooled immediately in liquid nitrogen.

**Data processing.** Single-crystal X-ray diffraction data for the BDBV223 apo-Fab or Fab-peptide complex were collected at beamline 23-ID-D of the Advanced Photon Source (Argonne National Labs, United States) or at beamline 12–2 of the Stanford Sychrotron Radiation Lightsource (SLAC National Accelerator Laboratory) respectively. Images were processed and scaled using XDS[25]. The initial model was determined by molecular replacement in Phaser[26]. Further model refinement procedures were carried out using Phenix.refine[27] and BUSTER software[28]. Iterative manual model building and correction were performed using COOT[29]. Apo-Fab and Fab-peptide complex structures have been deposited in the PDB under IDs 6N7U and 6N7J respectively.

**Mutagenesis.** cDNAs were synthesized (SGI-DNA Company) to encode the BDBV223 heavy chain with a A104F, I103A, I103E, S106A, or S106N mutation, or BDBV light chain with a Y33F mutation. Synthesized genes were cloned into a DNA plasmid expression vector encoding IgG1 and transformed into E. coli cells. Recombinant mAbs were produced after co-transfection of ExpiCHO cells (ThermoFisher Scientific) with respective DNA of mutant heavy chain and WT light chain, or WT heavy chain and mutant light chain using the manufacturer's protocol, and were purified as described above.

**GP expression and purification for binding ELISAs.** The ectodomains of EBOV GP ΔTM (residues 1–636; strain Makona; GenBank: KM233070), BDBV GP ΔTM (residues 1–643; strain 200706291 Uganda; GenBank: NC_014373), SUDV GP ΔTM (residues 1–637; strain Gulu; GenBank: NC_006432), and MARV GP ΔTM (residues 1–648; strain Angola2005; GenBank: DQ447653) were expressed transiently in Expi293F cells with a C-terminal strep II tag using the pcDNA3 plasmid vector. Secreted proteins were purified using 5 mL StrepTrap HP column (GE Healthcare) following the manufacturer's protocol, and then purified further and buffer exchanged into PBS using Superdex200 (GE Healthcare) size exclusion chromatography.

**ELISA.** Wells of microtiter plates were coated with purified, recombinant EBOV, BDBV, SUDV, or MARV GP ΔTM and incubated at 4 °C overnight. Plates were blocked with 2% non-fat dry milk and 2% normal goat serum in DPBS containing 0.05% Tween-20 (DPBS-T) for 1 h. Serial dilutions of mAb were applied to the wells and incubated for 1 h at ambient temperature. The bound antibodies were

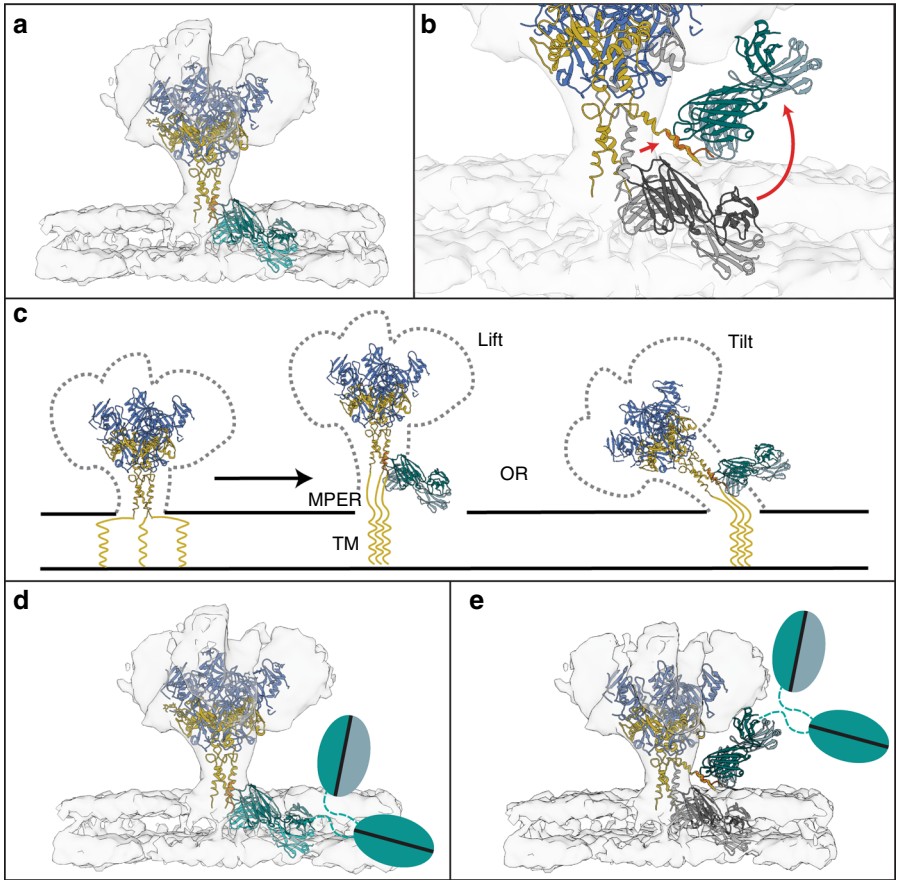

**Fig. 4** Necessary displacement of the stalk to achieve binding of BDBV223. **a** When the BDBV223-GP peptide complex structure is aligned to the CryoEM structure of membrane-anchored EBOV GP [EMD-8630][20] and the stalk-containing crystal structure of unbound EBOV GP [PDB: 5JQ7][16, 20], it is clear that the angle of approach of the BDBV223 Fab would clash with the expected position of the plasma membrane. **b, c** This steric hindrance indicates that the stalk regions of the GP may be (**b**) splayed from center upon antibody binding (gray to colored), or (**c**) lifted from membrane/tilted in order to expose the binding site. **d, e** Steric interference would be greater for the intact IgG of the wild-type antibody in either the unaltered (**d**) or splayed (**e**) conformation. Color Scheme: RGB

detected using goat anti-human IgG conjugated with HRP (diluted 1:5000) (Southern Biotech, CAT: 2040–05) and TMB substrate (ThermoFisher Scientific). Color development was monitored, 1 N hydrochloric acid was added to stop the reaction, and the absorbance was measured at 450 nm using a spectrophotometer (Biotek). $EC_{50}$ values for mAb binding were determined using Prism 7.2 software (GraphPad) after log transformation of antibody concentration using sigmoidal dose-response nonlinear regression analysis.

**pH ELISAs.** Wells of microtiter plates were coated with purified, recombinant EBOV and BDBV GP ΔTM and incubated at 4 °C overnight. Plates were blocked with 3% BSA in TBS for 1 h. Serial dilutions of BDBV223 were applied to the wells and incubated for 1 h at ambient temperature. BSA (Sigma-Aldrich) was re-solubilized and mAb was diluted with TBS at different pH ranging from 4.5–7.4. The bound antibodies were detected using goat anti-human IgG (diluted 1:5000) conjugated with HRP (Southern Biotech, CAT: 2040–05) and TMB substrate (ThermoFisher Scientific). Color development was monitored, 1 N sulfuric acid was added to stop the reaction, and the absorbance was measured at 450 nm using a Spark plate reader (Tecan Life Sciences). Prism 7.2 software (GraphPad) was used to determine the fluctuations of ELISA signal across pH values.

**Virus neutralization assay.** The neutralization assay with filoviruses was performed in the biosafety level 4 facility of the Galveston National Laboratory, University of Texas Medical Branch. The recombinant EBOV strain Mayinga expressing enhanced green fluorescent protein (eGFP) from an added gene[30] and its derivatives in which the GP was replaced with its counterpart from BDBV (strain 200706291 Uganda) or SUDV (strain 200011676 Gulu) were used[31], and referred to as EBOV, BDBV, and SUDV for simplicity. The assay was performed in a high-throughput format, as previously described[31].

**Somatic hypermutation.** The identities of gene segments and mutations from germline were determined by alignment using the ImMunoGeneTics database[32].

**Reporting Summary.** Further information on experimental design is available in the Nature Research Reporting Summary linked to this article.

## Data availability
Coordinates and structure factors have been deposited into the Protein Data Bank under accession numbers 6N7J and 6N7U for the complexed and apo BDBV223 structures, respectively. All other data are available from the author upon reasonable request.

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

## Acknowledgements

We acknowledge support by NIH/NIAID U19 109762 (EOS), U19 AI109711 (to J.E.C. and A.B.), Defense Threat Reduction Agency grant HDTRA1-13-1-0034 (to J.E.C. and A.B.), and NIH/NIAID F30 AI136410 (L.B.K.). This research used resources of the Advanced Photon Source, a U.S. Department of Energy (DOE) Office of Science User Facility operated for the DOE Office of Science by Argonne National Laboratory under Contract No. DE-AC02-06CH11357. Use of the Stanford Synchrotron Radiation Lightsource, SLAC National Accelerator Laboratory, is supported by the U.S. Department of Energy, Office of Science, Office of Basic Energy Sciences under Contract No. DE-AC-02-76SF00515. The SSRL Structural Molecular Biology Program is supported by the DOE Office of Biological and Environmental Research, and by the National Institutes of Health, National Institute of General Medical Sciences (including P41GM103393). The contents of this publication are solely the responsibility of the authors and do not necessarily represent the official views of NIGMS or NIH. We thank Sharon Schendel for manuscript editing. This is manuscript # 29723 of The Scripps Research Institute.

## Author contributions

Conceptualization by L.B.K., B.R.W., C.L.M., P.G., A.F., A.B., J.E.C., and E.O.S. Methodology by L.B.K., B.R.W., C.L.M., P.G., A.F., R.B., S.H., P.A.I., and K.H. Investigation by L.B.K., B.R.W., C.L.M., P.G., A.F., P.A.I., and K.H. Resources by A.B., J.E.C., and E.O.S. Writing original draft by L.B.K., B.R.W., and E.O.S. Writing, review, and editing by all authors. Visualization by L.B.K., B.R.W., C.L.M., and E.O.S. Supervision by A.B., J.E.C., and E.O.S. Project administration by A.B., J.E.C., and E.O.S. Funding acquisition by A.B., J.E.C., and E.O.S.

## Additional information

**Competing interests:** J.E.C. has served as a consultant for Takeda Vaccines, Sanofi Pasteur, Pfizer, and Novavax, is on the Scientific Advisory Boards of CompuVax, GigaGen, Meissa Vaccines, PaxVax, and is Founder of IDBiologics, Inc. The remaining authors declare no competing interests.

