## [Peer Review File · Nature Communications]

Reviewers' Comments:

Reviewer #1:

Remarks to the Author:

In the manuscript "Cross-reactive, neutralizing human survivor monoclonal antibody BDBV223 targets the ebolavirus stalk", King et al describe the crystal structure of antibody BDBV223 bound to a peptide fragment of its ebolavirus GP epitope. Antibody BDBV223 is of interest due to its cross-neutralizing reactivity with both BDBV and EBOV GP and its targeting of the stalk/membrane proximal external region (MPER) of GP2, a poorly understood region of the glycoprotein. As such, the structure delineates the molecular basis for neutralizing antibody recognition of this region of the glycoprotein with promising ramifications for downstream therapeutics and vaccine development.

Although BDBV223 is able to bind the soluble ectodomain of SUDV GP, it does not neutralize the SUDV virus. Its lack of SUDV neutralization has been partially attributed to a D624N polymorphism in the SUDV epitope. Attempts in the current study to use the structure to introduce mutations into BDBV223 that enable SUDV neutralization were apparently unsuccessful. The authors suggest factors outside the crystallized epitope in SUDV may contribute to lack of BDBV223 neutralization of this species.

Comments:

1. Neutralizing antibodies against HIV that recognize the membrane proximal external region (MPER) of gp41 require both binding to the glycoprotein and interactions with the viral membrane. Given the proximity of the BDBV223 epitope to the viral membrane, is there structural evidence for membrane-binding motifs in the BDBV223 antibody? For instance, hydrophobic patches or basic patches that might interact with lipid headgroups?
2. If the antibody acts by preventing conformational transition to six helix bundle, it must do so within acidic endosomes. How does acidic pH influence BDBV223 recognition of the GPs, and are there differences between species?
3. BDBV223 is suggested to interfere with glycoprotein rearrangement involved in membrane fusion. Mapping of the BDBV223 epitope in the context of postfusion six helix bundle structure would allow for direct visualization of degree of compatibility of postfusion conformation with BDBV223 binding.
4. Superposition of the bound peptide onto the structure of EBOV GP (5JQ7) is used to orient the antibody-peptide complex relative to rest of the GP ectodomain and to reveal antibody clashes with neighboring trimer stalk. However, the 5JQ7 structure of EBOV GP has a fibrin foldon trimerization motif fused onto its C-terminus. This trimerization domain may or may not impose non-native structural constraints on the stalk region. As such, are there EM maps of this region that reflect an unconstrained stalk region that can be used for orienting the peptide-bound antibody structure in the native spike?
5. Based on the structure, a set of mutations is introduced into BDBV223 to try to recover SUDV neutralization. The positions of these mutations in the structure and the rationale behind them should be described more fully. Given the significance of BDBV223 residue R100 in mediating a salt bridge with D624 in BDBV, it seems mutagenesis of this antibody residue would be relevant, especially in the context of the N624 polymorphism in SUDV.
6. What is the extent of somatic hypermutation of BDBV223? A sequence alignment showing mature antibody heavy and light chains aligned against germline precursors with residues that interact with

GP2 specified would be helpful.

7. Described structural differences between the apo and bound forms of the antibody are not linked to any functional ramifications for the antibody and as such are overly emphasized.

8. Are there viral escape mutations of BDBV223 and do they fall within epitope? If so, does structure inform on how they may lead to virus escape?

Minor Comments:

1. A sequence alignment across ebolaviruses of the BDBV223 epitope in GP2 with residues bound by the antibody shown is not currently included in the manuscript but would provide context in terms of sequence conservation and antibody cross-reactivity, as well as proximity to transmembrane domain.

2. Electron density map for most critical antibody-peptide interactions should be shown.

Reviewer #2:

Remarks to the Author:

King L. et al. describes the crystal structure of a neutralizing mAb isolated from a BDBV-survivor that targets the stalk region of the GP, in complex with a short peptide that represents its epitope. There is an interest in studying how such mAbs neutralize the viruses and the factors that govern their breadth. The main conclusion of this paper is that this mAb would only bind GP when the helical stalk region is separated from the trimeric bundle and emerges out from the membrane. Also, using the structural data the authors rationalize why this mAb can also target EBOV but not SUDV.

A major caveat of this work is the lack of data that support the correct modeling of the peptide at the very low-resolution achieved. At worse than 3.5Å, density is typically observed only for large bulky side-chains and this may lead to mistakes.

Specifically, the authors used a synthetic peptide made of 16 amino acids but modeled only 15 residues (and not as the author state in the text, see below). The missing density may belong to a residue at the C' terminus or at the N' terminus of the peptide. Understanding which of the options is correct is critical as it may completely alter the conclusions of the authors regarding the potential clash of the mAb with the nearby helices of the GP (Figure 1c), as well as the proposed interactions between the peptide and the mAb. To help convince that the peptide was indeed correctly modeled, the authors will need to show the two options, i.e. to model the peptide in the two possible registries. Then, a series of side-by-side close-up views of the all the different side chains of the peptide in both cases should be displayed in a simulated annealing composite omit map. There should be clear indications that would favor one of the modeling options over the other before this proposed interaction mode could be considered as correct.

Following on the low-resolution limitation, the authors describe an induced fit of the mAb when it binds to the peptide, including a 1.6Å shift in CDRH2, a rotameric rearrangement of Y53, and rearrangement of CDRH3. Again, all of these observations will need to be demonstrated by showing electron density that will support the modeling. The map for all of these figures will have to be a simulated annealing composite omit map. Likewise, all the putative interactions that are highlighted in Figure 3 will also need to be accompanied with figures showing the density of the relevant residues using simulated annealing composite omit map.

In the result section lines 226-227 the authors state, "The entire 620-635 peptide, thought to represent the BDBV223 epitope, is visible in the crystal structure." But this is not the case as the deposited structure to the PDB is missing residue 635. Not only this statement is incorrect, as

explained above this may have substantial consequences.

Additional comments:

- 1) Abstract - the mAb cannot "separate", "lift", or "bend" the stalk region. It can merely stabilize such conformations when are naturally sampled by the GP at some frequency, please correct the text.
- 2) Introduction & Results – Diffraction data extends to 3.68Å, rounding this number gives 3.7 and not as erroneously stated in the text.
- 3) Mapping the interactions between the peptide and the mAb seems too detailed to the level of information that a 3.7Å map can actually provide. Can the authors be completely sure that the distance of D624 from R100 is less than 4.0Å (assuming the entire modeling is correct)? Otherwise, this interaction is not a salt bridge. Less definitive language might be more appropriate.
- 4) The authors present binding data to various GPs (Figure S1). There are marked differences between the "hybridoma" produced mAb and the recombinant mAb that binds less strongly. This may indicate the presence of a PTM or inaccurate sequencing of the original mAb. The authors should try to follow on that using MS analysis of both proteins.

Cross-reactive, neutralizing human survivor monoclonal antibody BDBV223 targets the ebolavirus stalk

Response to reviewers:

Reviewer #1 (Remarks to the Author):

Comments:

1. Neutralizing antibodies against HIV that recognize the membrane proximal external region (MPER) of gp41 require both binding to the glycoprotein and interactions with the viral membrane. Given the proximity of the BDBV223 epitope to the viral membrane, is there structural evidence for membrane-binding motifs in the BDBV223 antibody? For instance, hydrophobic patches or basic patches that might interact with lipid headgroups?

Overall, the membrane-proximal surface of the Fab is less hydrophilic than the more solvent-exposed surface, but not dramatically so. There is one basic patch just beneath the paratope (primarily formed by residues from FRL3) that could theoretically interact with a phospholipid head group. We have included an additional supplemental figure showing the electrostatics map with this region highlighted. A short explanation has been added to the text as well.

2. If the antibody acts by preventing conformational transition to six helix bundle, it must do so within acidic endosomes. How does acidic pH influence BDBV223 recognition of the GPs, and are there differences between species?

This is a good question, and we thank the reviewer for raising it. We have performed additional binding experiments and find no difference in binding whether at neutral or acidic pH. We have included this data in the revised manuscript as Supplementary Figure 9. Relevant text has been added as well.

3. BDBV223 is suggested to interfere with glycoprotein rearrangement involved in membrane fusion. Mapping of the BDBV223 epitope in the context of postfusion six helix bundle structure would allow for direct visualization of degree of compatibility of postfusion conformation with BDBV223 binding.

This is another good suggestion. We have included this model in the revised manuscript as Supplementary Figure 8. BDBV223 does not appear to dramatically clash with the post-fusion structure itself, but could interfere sterically with the transition to this structure, particularly as the intact IgG.

4. Superposition of the bound peptide onto the structure of EBOV GP (5JQ7) is used to orient the antibody-peptide complex relative to rest of the GP ectodomain and to reveal antibody clashes with neighboring trimer stalk. However, the 5JQ7 structure of EBOV GP has a fibrin foldon trimerization motif fused onto its C-terminus. This trimerization domain may or may not impose non-native structural constraints on the stalk region. As such, are there EM maps of this region that reflect an unconstrained stalk region that can be used for orienting the peptide-bound antibody structure in the native spike?

This is an excellent point. Unfortunately none of the EM maps or other crystal structures of EBOV GP have strong enough density in the stalk region to offer a different model or create a better alignment. Our recent structure of marburgvirus GP, however, was not expressed with a foldon or any other trimerization domain (King, *et al.* 2018 *Cell Host and Microbe*). In this structure, the marburgvirus stalk is a tight three-helix bundle, like that of Ebola virus. We agree with the reviewer that the fibritin domain may impose non-native structural constraints on the stalk region; however, at this time we do not have any alternative information for ebolaviruses. We have added text clarifying this point.

5. Based on the structure, a set of mutations is introduced into BDBV223 to try to recover SUDV neutralization. The positions of these mutations in the structure and the rationale behind them should be described more fully. Given the significance of BDBV223 residue R100 in mediating a salt bridge with D624 in BDBV, it seems mutagenesis of this antibody residue would be relevant, especially in the context of the N624 polymorphism in SUDV.

Thank you for this suggestion. We have added additional text describing why certain mutations were chosen. Specifically, R100 was separately mutated to an alanine, isoleucine, and a tryptophan in an attempt to accommodate the N624 polymorphism. None of these mutant antibodies, however, bound to any of EBOV, SUDV, or BDBV GP. We have added text within the supplemental data clarifying this point.

6. What is the extent of somatic hypermutation of BDBV223? A sequence alignment showing mature antibody heavy and light chains aligned against germline precursors with residues that interact with GP2 specified would be helpful.

The extent of somatic hypermutation of the heavy and light chain is 15.5% and 14.7% respectively. We have added the sequence alignment as Supplementary Figure 2 and have included relevant text in the revised manuscript.

7. Described structural differences between the apo and bound forms of the antibody are not linked to any functional ramifications for the antibody and as such are overly emphasized.

We agree that at this point we do not currently know a functional ramification beyond that induced fit in multiple CDRs seems to occur upon binding. We have condensed the text describing these structural changes, but feel they should still be noted in case future studies link presence or absence of structural rearrangement to potency of neutralization or success in cross-reactivity.

8. Are there viral escape mutations of BDBV223 and do they fall within epitope? If so, does structure inform on how they may lead to virus escape?

This is an excellent point. There is one published escape mutant that falls within the epitope of BDBV223. The mutant, P634H, could interfere with the binding of BDBV223 in several ways, but the two we believe are most important relate to sterics and secondary structure. The primary clash that would arise with this mutation would be between P634H and R94

of CDR-L3. We have added text addressing this escape mutant and the possible ways in which it could lead to viral escape.

Minor Comments:

1. A sequence alignment across ebolaviruses of the BDBV223 epitope in GP2 with residues bound by the antibody shown is not currently included in the manuscript but would provide context in terms of sequence conservation and antibody cross-reactivity, as well as proximity to transmembrane domain.

This figure has been included in the revised manuscript as Supplementary Figure 1.

2. Electron density map for most critical antibody-peptide interactions should be shown.

These maps have been included in the revised manuscript as Supplementary Figures 4 and 5.

We thank the reviewer for their comments and suggestions, and think that his or her suggestions have improved the study.

Reviewer #2 (Remarks to the Author):

King L. et al. describes the crystal structure of a neutralizing mAb isolated from a BDBV-survivor that targets the stalk region of the GP, in complex with a short peptide that represents its epitope. There is an interest in studying how such mAbs neutralize the viruses and the factors that govern their breadth. The main conclusion of this paper is that this mAb would

only bind GP when the helical stalk region is separated from the trimeric bundle and emerges out from the membrane. Also, using the structural data the authors rationalize why this mAb can also target EBOV but not SUDV.

A major caveat of this work is the lack of data that support the correct modeling of the peptide at the very low-resolution achieved. At worse than 3.5Å, density is typically observed only for large bulky side-chains and this may lead to mistakes.

Specifically, the authors used a synthetic peptide made of 16 amino acids but modeled only 15 residues (and not as the author state in the text, see below). The missing density may belong to a residue at the C' terminus or at the N' terminus of the peptide. Understanding which of the options is correct is critical as it may completely alter the conclusions of the authors regarding the potential clash of the mAb with the nearby helices of the GP (Figure 1c), as well as the proposed interactions between the peptide and the mAb. To help convince that the peptide was indeed correctly modeled, the authors will need to show the two options, i.e. to model the peptide in the two possible registries. Then, a series of side-by-side close-up views of the all the different side chains of the peptide in both cases should be displayed in a simulated annealing composite omit map. There should be clear indications that would favor one of the modeling options over the other before this proposed interaction mode could be considered as correct.

We have added two supplemental figures (4 and 5) that illustrate simulated annealing composite omit maps that indicate clearly that the presented amino acid register is correct. If we attempt to model the peptide in any alternate register, two large aromatic residues within the peptide (H628 and F630) move from entirely within good electron density to

almost entirely outside of the electron density. We have included close up views that specifically illustrate these residues in Supplementary Figure 4.

Following on the low-resolution limitation, the authors describe an induced fit of the mAb when it binds to the peptide, including a 1.6Å shift in CDRH2, a rotameric rearrangement of Y53, and rearrangement of CDRH3. Again, all of these observations will need to be demonstrated by showing electron density that will support the modeling. The map for all of these figures will have to be a simulated annealing composite omit map. Likewise, all the putative interactions that are highlighted in Figure 3 will also need to be accompanied with figures showing the density of the relevant residues using simulated annealing composite omit map.

In the revised manuscript, we have included simulated annealing composite omit maps of each of the CDRs in the peptide bound and apo-structures and interactions that they form in Supplementary Figures 4 and 5. All of the maps confirm the conformations of CDRs illustrated in the main text Figures 2 and 3.

In the result section lines 226-227 the authors state, “The entire 620-635 peptide, thought to represent the BDBV223 epitope, is visible in the crystal structure.” But this is not the case as the deposited structure to the PDB is missing residue 635. Not only this statement is incorrect, as explained above this may have substantial consequences.

We have edited the statement to reflect that residues 620-634 are visible in the structure.

Additional comments:

1) Abstract - the mAb cannot “separate”, “lift”, or “bend” the stalk region. It can merely stabilize such conformations when are naturally sampled by the GP at some frequency, please correct the text.

This change has been made in the text.

2) Introduction & Results – Diffraction data extends to 3.68Å, rounding this number gives 3.7 and not as erroneously stated in the text.

This change has been made in the text.

3) Mapping the interactions between the peptide and the mAb seems too detailed to the level of information that a 3.7Å map can actually provide. Can the authors be completely sure that the distance of D624 from R100 is less than 4.0Å (assuming the entire modeling is correct)? Otherwise, this interaction is not a salt bridge. Less definitive language might be more appropriate. **This is a good question. In this case, the electron density at the site of the salt bridge is extremely clear and B values of the interacting residues are 40 (D624) and 57 (R100) (vs 54 for the entire structure). The clarity of the map in this region indicates that the bridging atoms are less than 4 Å apart (3.8 Å), and that the interaction is most likely a salt bridge. We have changed the text to use less definitive language.**

4) The authors present binding data to various GPs (Figure S1). There are marked differences between the “hybridoma” produced mAb and the recombinant mAb that binds less strongly. This

may indicate the presence of a PTM or inaccurate sequencing of the original mAb. The authors should try to follow on that using MS analysis of both proteins.

We agree with the reviewer that this is an interesting question. Sequences for both chains were identified by 5-prime RACE approach and from multiple batches of clonal hybridoma cells and were confirmed to be identical. This sequencing approach eliminates potential errors in framework regions 1 and 4 observed when framework region-embedded primers used for sequencing. The constructs used for transfection were sequence-verified as well. Overall, it is common (usual) to see different activity patterns for antibodies expressed in hybridoma, vs. CHO, vs 293F. This is well known in the industry. One other thought is that antibody isotype may play a role in the difference. WT hybridoma mAb is of IgG3, and recombinant mAb presented in supplemental of current manuscript is of IgG1 isotype. However, we did perform side-by-side comparisons of hybridoma IgG3, recombinant IgG1, recombinant IgG3, as well as recombinant IgG1 LALA. The results indicate that binding of recombinant IgG3 is only slightly different for SUDV GP binding. but identical for BDBV and EBOV GPs when compared to WT hybridoma antibody. We find that IgG1, whether WT or LALA, bound with lesser extent to EBOV and SUDV when compared to either hybridoma or recombinant IgG3s. Again, all sequences were confirmed by 5' RACE. We are happy to include this information as a supplemental figure, although it is not necessarily a focus of this paper. We leave this up to the discretion of the editor.

Reviewers' Comments:

Reviewer #1:

Remarks to the Author:

In the revised version of the manuscript "Cross-reactive, neutralizing human survivor monoclonal antibody BDBV223 targets the ebolavirus stalk", King et al address many concerns raised previously by the reviewers. In the revised manuscript they now include electron density maps, analysis of surface electrostatics, compatibility with post-fusion six-helix bundle, and binding studies at low pH, all of which improve the manuscript. New figures and data associated with these revisions are placed however into the supplemental information section, leaving main text figures largely unmodified.

Comments:

1. One of the main postulates of the paper, that remains unchanged, is that binding of the BDBV223 antibody to its stalk epitope must be to a state of the stalk that is either splayed, lifted, or tilted so as to avoid clashes with neighboring trimer stalk helices or with the viral membrane. This conclusion is based on docking of the BDBV223 complex structure onto an EBOV GP crystal structure (5JQ7) that fits into an EM map of a native virion GP. A potential problem, as noted previously, is that the 5JQ7 structure has a fibritin foldon motif at its C-terminus and this may impose artificial constraints on the helical bundle stalk. The antibody clashes seen with a neighboring stalk helix (Fig. 1c) could therefore be due to foldon-imposed constraints that tighten the helical bundle. It is unclear if such clashes would also occur in a more native unconstrained helical bundle without foldon.

In their reply and revised text the authors note that a recent structure of marburg GP without foldon reveals an extended stalk with similar architecture. A direct comparison of this unconstrained marburg stalk vs. the foldon-constrained ebola stalk (albeit of different viruses) could be informative, especially in regards to the diameters of the respective helical bundles at their C-terminal ends. Superposition of the structure of BDBV223 bound to peptide onto the marburg GP stalk (although not its natural target) could also help assess if clashes are observed with neighboring stalk helices in an unconstrained setting.

2. Described structural differences between the apo and bound forms of the antibody provide little if any insight into function but are still placed prominently as Figure 2 in the revised text – distracting from more functionally relevant findings. Though it is possible that at some point in the future such differences may be shown to have implications for potency of neutralization, these implications are not part of the current study nor are they discussed.

3. Although description of the light chain of BDBV223 is invoked in several places in the text, these descriptions are not reflected in the figures. For instance, lines 157-161 in the text describe interactions of the light chain CDRL1 and CDRL3 with the peptide, but there is no figure that shows these interactions. The P634H escape mutation is suggested to be due to a clash with the light chain residue R94 of the CDRL3, but there is no figure that describes this interaction.

4. The apo structure of BDBV223 is of higher resolution than the complex (2.0Å vs 3.7Å), but refinement of the apo structure has inferior statistics. Rfree values are high at 27% for a 2Å structure (compared to 25.4% for 3.7Å complex). The validation report also shows a number of outliers including in planarity and bond angles. It also shows many backbone and sidechain outliers, unexpected for a 2Å structure (e.g., 13 rotameric outliers).

5. Figure S2 sequence alignments do not show residue numbers nor do they show residues that contact the peptide epitope. Likewise, Figure S1 alignment does not show residues that mediate contacts with BDBV223.

6. Line 97 of text states that resolution of apo structure is 2.2Å, but Supplemental Table 2 suggests that it is 2.03Å.

7. Overall, graphical representations of the structures in the figures do not do justice to the structural findings and in many cases are difficult to decipher.

Reviewer #2:

Remarks to the Author:

In the revised version of their manuscript the authors addressed my previous comments but some minor issues remain.

The authors are convinced in their modeling of the peptide and they provide figure S4 to illustrate that.

Compared to the very informative figures for the CDRs (figure S3), the figure for the peptide is less persuading and need to be further improved.

Density maps in panels 'C' and 'F' should be identical as expected from SA-composite omit maps and as stated in the figure legend but in fact show some puzzling differences (see upper density blob in the blue vs. green maps). The authors should explain that or somehow fix the maps (make sure the same sigma level is used or maybe use Fo-Fc map after omitting the peptide instead).

Also, there is an inherent limitation in evaluating fit to electron density using 2D figures. As a service to the future readers of this manuscript the authors should further provide simple supplementary movies showing both models in the SA-composite omit maps while rotating them.

Responses to reviewers are written in blue below.

Reviewer #1 (Remarks to the Author):

Comments:

1. One of the main postulates of the paper, that remains unchanged, is that binding of the BDBV223 antibody to its stalk epitope must be to a state of the stalk that is either splayed, lifted, or tilted so as to avoid clashes with neighboring trimer stalk helices or with the viral membrane. This conclusion is based on docking of the BDBV223 complex structure onto an EBOV GP crystal structure (5JQ7) that is fit into an EM map of a native virion GP. A potential problem, as noted previously, is that the 5JQ7 structure has a fibrin foldon motif at its C-terminus and this may impose artificial constraints on the helical bundle stalk. The antibody clashes seen with a neighboring stalk helix (Fig. 1c) could therefore be due to foldon-imposed constraints that tighten the helical bundle. It is unclear if such clashes would also occur in a more native unconstrained helical bundle without foldon.

In their reply and revised text the authors note that a recent structure of marburg GP without foldon reveals an extended stalk with similar architecture. A direct comparison of this unconstrained marburg stalk vs. the foldon-constrained ebola stalk (albeit of different viruses) could be informative, especially in regards to the diameters of the respective helical bundles at their C-terminal ends. Superposition of the structure of BDBV223 bound to peptide onto the marburg GP stalk (although not its natural target) could also help assess if clashes are observed with neighboring stalk helices in an unconstrained setting.

We have included a supplemental figure (Supplemental Figure 8) that displays an alignment of BDBV223 to a Marburgvirus GP structure (PDB: 6BP2), which does not include a trimerization domain in the construct. We would also like to point out that marburgviruses and ebolaviruses are quite sequence divergent in this area (and in general). Despite these differences, this figure shows the overall structure of the helices aligning well. Possibly due to the absence of a trimerization domain, the C termini of the marburgvirus helices splay out 7.7 Å wider than the ebolavirus helices. However, even with this difference in spacing, a clash still occurs in both ebolavirus and marburgvirus models. Our modeling continues to support the notion that ebolavirus GP would have to splay/tilt in order to accommodate the binding of BDBV223 to avoid interacting with the membrane (see Fig 4).

2. Described structural differences between the apo and bound forms of the antibody provide little if any insight into function but are still placed prominently as Figure 2 in the revised text – distracting from more functionally relevant findings. Though it is possible that at some point in the future such differences may be shown to have implications for potency of neutralization, these implications are not part of the current study nor are they discussed.

We have added additional text to this section to further discuss the degree of high conformational flexibility in this region. We believe that these large changes may be important to consider in understanding how antibodies may recognize this key region and be elicited by

vaccination strategies. They likely affect binding energy and have influences how germline antibodies and somatically rearranged antibodies evolved to recognize this region. These structural rearrangements should be considered in future mutagenesis experiments to improve the binding of BDBV223, and may be more broadly applicable to the growing field of antibody engineering. As this figure represents a body of experimental data not otherwise available, we would like to keep this as a main figure, but will leave it to the discretion of the editor as to whether it should be included in the supplemental or main text.

3. Although description of the light chain of BDBV223 is invoked in several places in the text, these descriptions are not reflected in the figures. For instance, lines 157-161 in the text describe interactions of the light chain CDRL1 and CDRL3 with the peptide, but there is no figure that shows these interactions. The P634H escape mutation is suggested to be due to a clash with the light chain residue R94 of the CDRL3, but there is no figure that describes this interaction.

We have added supplementary figure panels illuminating interactions of CDRs L1 and L3 with GP, and suggest that a P634H escape mutation could introduce a clash with R94.

4. The apo structure of BDBV223 is of higher resolution than the complex (2.0Å vs 3.7Å), but refinement of the apo structure has inferior statistics. Rfree values are high at 27% for a 2Å structure (compared to 25.4% for 3.7Å complex). The validation report also shows a number of outliers including in planarity and bond angles. It also shows many backbone and sidechain outliers, unexpected for a 2Å structure (e.g., 13 rotameric outliers).

Thank you for bringing this to our attention. Those were preliminary tables, and have now been replaced with the final tables (PDB accession codes 6N7U (apo) and 6N7J (complex)). We have included the final validation reports for these models with our uploaded files.

5. Figure S2 sequence alignments do not show residue numbers nor do they show residues that contact the peptide epitope. Likewise, Figure S1 alignment does not show residues that mediate contacts with BDBV223.

We have added these features to Figures S1 and S2. Further, we have added a sequence alignment table (Supplemental Table 3) to aid in aligning with the IMGT antibody numbering scheme.

6. Line 97 of text states that resolution of apo structure is 2.2Å, but Supplemental Table 2 suggests that it is 2.03Å.

Thank you for pointing out this inconsistency. The text has been corrected to state the resolution properly as 2.03 Å.

7. Overall, graphical representations of the structures in the figures do not do justice to the structural findings and in many cases are difficult to decipher.

Thank you for this helpful critique. We have redesigned several figures to improve clarity, as follows:

-Updated Figure 1A to more clearly display the peptide binding to the BDBV223 Fab including a top-down view

-New figure 1B for orienting the reader to the CDRs in relation to the peptide.

-New Figure 3D and E showing the relationship of the peptide to L1 and L3 as well as the modeled escape mutant

-Updated electron density figure S4 to improve clarity and consistency of the mesh between panels.

-Uploaded a video to help with 3-dimensional visualization of the peptide (and the less likely alternative register) in the density

We thank the reviewer for their helpful comments in improving this manuscript.

Reviewer #2 (Remarks to the Author):

In the revised version of their manuscript the authors addressed my previous comments but some minor issues remain.

The authors are convinced in their modeling of the peptide and they provide figure S4 to illustrate that.

Compared to the very informative figures for the CDRs (figure S3), the figure for the peptide is less persuading and need to be further improved.

Density maps in panels 'C' and 'F' should be identical as expected from SA-composite omit maps and as stated in the figure legend but in fact show some puzzling differences (see upper density blob in the blue vs. green maps). The authors should explain that or somehow fix the maps (make sure the same sigma level is used or maybe use Fo-Fc map after omitting the peptide instead).

We have updated figure S4 to correct small discrepancies in the rendering of the density mesh. We found that this discrepancy was due to a glitch during importation of the density map into PyMol. We have replaced the figure entirely to help make the results more clear. We have also included explanations of how we generated the maps in the figure legend: a SA-composite omit map was generated for a model with the peptide deleted. We then included both potential peptide registers and evaluated their fit. We also ran a refinement on these models and have included the results of those as well showing Fo-Fc maps that clearly favor the published register over the other.

Also, there is an inherent limitation in evaluating fit to electron density using 2D figures. As a service to the future readers of this manuscript the authors should further provide simple

supplementary movies showing both models in the SA-composite omit maps while rotating them.

We have uploaded a short movie that rotates around the peptide in the SA-composite omit maps. The movie shows both registers exactly as displayed in the corresponding supplemental figure (Figure S4).

We thank the reviewer for their helpful comments in improving this manuscript.

Reviewers' Comments:

Reviewer #1:

Remarks to the Author:

The authors have addressed all issues raised previously and I support publication.

Reviewer #2:

Remarks to the Author:

The authors have addressed all of my comments.